# Adsorption Behaviors of a Twin-Tower Hydrogen Purification System Mounted onto Staggered Stainless Steel Sheets Coated with Composite Membrane

**DOI:** 10.3390/membranes11030169

**Published:** 2021-02-27

**Authors:** Hung-Ta Wu, Chin-Chun Chung

**Affiliations:** 1Department of Chemical and Materials Engineering, National Ilan University, No.1, Sec. 1, Shennong Rd., Yilan City, Yilan County 260007, Taiwan; 2Department of Chemical Engineering, Army Academy, No.750, Longdong Rd., Chung-Li District, Taoyuan City 320316, Taiwan; doctorjons@aaroc.edu.tw

**Keywords:** adsorption, activated carbon, membrane, switching time

## Abstract

Many studies have been conducted on hydrogen production, storage, purification, and transportation. The use of fixed-bed adsorption towers for hydrogen purification is common. The operating variables involved that could affect the adsorption behavior, such as the amount of adsorbents used, the flow rate, and the concentration of the adsorbate, should be discussed further. In addition, the pressure drop caused by the operation of the adsorption tower still needs to be considered. Therefore, the staggered stainless steel sheet coatings with SiO_2_/MCM41/activated carbon composite membrane were mounted in a twin-tower adsorption system to purify the hydrogen. Similar to the pressure swing adsorption (PSA) system, the amounts of SiO_2_, activated carbon, and molecular sieves used in the adsorption tower were changed into the amounts of tetraethoxysilane (TEOS), activated carbon powder, and MCM41 powder added to the casting solution. The experimental results showed that the performance of this twin-tower hydrogen purification system would not be increased when one of the target adsorbents was excessive. In addition, the outflow of non-hydrogen components was found to be early when a certain adsorbent was not sufficient. Finally, the recommended switching time for this system was set at an adsorption capacity reaching about 75% saturated capacity.

## 1. Introduction

Hydrogen energy may become an important secondary energy in the 21st century due to its higher calorific value. For example, the calorie value after burning hydrogen is about 3 times that of gasoline, 3.9 times that of alcohol, and 4.5 times that of coke. Hydrogen is the cleanest energy in the world because the product of the reaction between hydrogen and oxygen is water only. The chemical methods for hydrogen production include steam reformation, partial oxidation, water gas reaction, electrolysis, and photocell reaction. In addition to these chemical methods, industrial exhaust was also used as the feed, and physical methods such as pressure swing adsorption (PSA) and desorption were used to obtain extremely pure hydrogen at the end of the treated system for recycling. The principle adsorbents with different adsorption abilities for various gas molecules were applied to achieve gas purification. The concept of intermolecular affinity was used by the PSA hydrogen purification method to recover and produce high-purity hydrogen. Then, the pressure was adjusted to desorb the impurities to facilitate the regeneration and recycling of the adsorbent. In general, there are three layers, including alumina and silica gel to adsorb water; activated carbon to adsorb CO_2_ and hydrocarbons; and a molecular sieve to adsorb O_2_, N_2_, CO, and CH_4_, in the fixed-bed hydrogen purification system. A previous study [1] showed that carbon dioxide could be removed successfully by an adsorber with a number of staggered alumina coated with a molecular sieve membrane. Compared with a traditional packed-bed adsorber, the lower pressure drop is for the adsorber packed by a number of staggered stainless steel sheets. Therefore, the adsorber packed by the staggered stainless steel sheets coated with a SiO_2_/MCM41/activated carbon composite membrane was used to purify hydrogen. In addition, the staggered arrangement in the adsorption bed also provides a longer contact time, making the adsorption capability equivalent to that of the traditional fixed-bed adsorber.

Rapid pressure swing adsorption (RPSA) can be tracked back to the 1970s. Kowler et al. [2] used fine particles as the adsorbent in a single-tower adsorption system, and its life cycle was about 2.5~20 s, which could be regarded as the predecessor of RPSA. Tamura [3] then proposed that the flushing process with the same direction replaced the decompression process with the same direction, thereby improving the purity of the strongly adsorbed material at the end of the tube. There are usually two flushing steps during the PSA process: one is to flush the strongly adsorbed substance out by high pressure, and the other is to flush the weakly adsorbed substance out by low pressure, so that two adsorbates can be obtained. This concept was used by Sircar and Zondolo [4] to obtain high-purity oxygen and nitrogen at the same time. In general, the purification of hydrogen could be accomplished by an adsorption tower or a membrane separation procedure. For example, Yang and Doong [5] used activated carbon in a fixed-bed adsorber to separate H_2_ and CH_4_. Multiple zeolite adsorbent layers were provided by Watson and Whitley [6] to construct a set of industrial PSA systems for gas separation. Recently, to obtain hydrogen and carbon dioxide at high levels using a reference configuration and by implementing PSA in natural gas combined cycle (NGCC) power plants, Solares and Wood [7] used a laboratory four-bed and seven-step PSA model to operate in an NGCC power plant. The levels of 95.37% carbon dioxide and 99.99% hydrogen were obtained with a purge to feed flow ratio of 0.22. Furthermore, to increase the adsorption rate of carbon dioxide and the purification of hydrogen, He et al. [8] adopted the dip-coating method to prepare a structured activated carbon adsorbent to mount inside an RPSA system. The results showed a higher mass transfer coefficient and a shorter mass transfer zone length for the structured activated carbon adsorbent in the RPSA system. The difference between literature studies and this study is that the membrane was used in the adsorber to purify hydrogen from multiple components.

For membrane separation, the versatility of PEEK-WC, which is an amorphous modified poly(ether ether ketone) with cardo groups, asymmetric membranes created industrial interest, thus Brunetti et al. [9] used a wet-phase inversion technique to prepare this membrane. The better selectivity for H_2_ than for N_2_ and CH_4_ was proven by gas separation tests. To reduce CO_2_ emissions, Chen et al. [10] argued that a Pebax/poly(dimethylsiloxane)/polyacrylonitrile composite hollow-fiber membrane was worth developing for flue gas treatment and hydrogen purification. The higher gas permeability and selectivity could be attributed to the fact that the concepts of the PDMS gutter layer and the interaction between poly(ethylene oxide) and CO_2_ were used in the study. Since a deposition of a thin palladium layer could increase the hydrogen selectivity, the vacuum electroless plating technique (VELP) was used by Kim et al. [11] to deposit Pd onto surface-treated polybenzimidazole. Mechanical abrasion and wet chemical and O_2_ plasma treatment were applied to increase the number of Pd anchoring sites and the hydrophilicity of the PBI-HFA surface. The enhancement of the H_2_ permselectivity of the Pd/PBI-HFA composite membrane treated by O_2_ plasma was demonstrated by the permeation measurement of the mixture of H_2_, N_2_, CO_2_, and CO. The roughness of the composite membranes was improved to deposit a thin metal layer well. In addition, just two Pd plating cycles were needed after the graphite coating to obtain infinite H_2_/N_2_ selectivity. Recently, Itta et al. [12] adopted the dry/wet inversion method to prepare a polyethermide-derived CMS membrane for the separation of H_2_ and N_2_, and the results demonstrated that the optimum H_2_/N_2_ permselectivity could be obtained from a CMS membrane fabricated with the dry-phase method and pyrolysis at 600 °C. To improve the long time lag and the membrane deactivation in purifying hydrogen, Budhi et al. [13] used a Pd-based membrane, such as Pd/a-Al_2_O_3_ membrane, and adopted the method of dynamic membrane operation to show that the recovery of hydrogen was increased with a decrease in the feed gas amplitude. The recovery of hydrogen reached 63% operating at a 300 s switching time and a 0.5 mL flow rate amplitude. Although a high level of hydrogen purification can be accomplished by both adsorption and membrane separation procedures, the larger treated amount should be implemented by an adsorption system. Furthermore, data relating to the mixtures of H_2_, CO_2_, N_2_, and H_2_O in flue gas are rare, so the SiO_2_/MCM41/activated carbon composite membrane was prepared and the staggered stainless steel sheet coated onto the membrane was adopted in this study. Therefore, the purpose of this study is to use the SiO_2_/MCM41/activated carbon composite membrane in a staggered adsorber to purify H_2_ from mixed gas.

## 2. Experimental Section

### 2.1. Preparation of Composite Membrane

Mixtures of tetraethoxysilane (TEOS), 0.79 mL ethanol, and 1.25 mL water were prepared and the pH of the mixture was adjusted to 2.0 by 0.1 M HCl. After stirring for one hour, MCM41 (purchased from china nano com) and activated carbon made from coconut shell (purchased from Effigen Carbon SDN. BHD) were added. Compared with other molecular sieves, the adsorption of water vapor was worse for the MCM41, so it was added to prepare the casting solution to reduce the adsorption of water vapor but focus on the adsorption of nitrogen. Since some studies [14,15] showed that carbon dioxide could be adsorbed by the activated carbon fabricated from coconut shell, this kind of activated carbon was chosen to be added into the casting solution. Although the sol-gel process might be complete within 2–4 h [16,17], the stirring time for the casting solution might take half a day after adding MCM 41 [18,19]. Therefore, the mixture was stirred at room temperature for 12 h. After the mixed solution was stirred evenly, the solution was put into a vacuum oven to remove air bubbles. Then, it was placed at room temperature for 24 h to return to the environmental temperature. Similar to the previous study, a stainless steel sheet was used as the carrier, and the size of the carrier was 18 × 18 × 2 mm. The carrier was immersed in the solution at room temperature, and then it was withdrawn at a speed of 1 mm/sec. The carrier covered with the mixed solution was calcined, and the heating rate was maintained at a speed of 2 °C/min up to 400 °C. The covered carrier was maintained at 400 °C for two hours, and then was placed in the normal condition for one day to return to the room temperature. These coating steps should be repeated three times to form a membrane with a thickness of about 0.2 cm.

### 2.2. Adsorber and Adsorption Test

A comparison between the fixed-bed adsorber and the tower with staggered stainless steel coated with the composite membrane shows a lower pressure drop for the latter. To compare the pressure drop between them, the heights of the fixed-bed adsorber at 20 cm and the tower with 20 staggered stainless steel pieces coated with the composite membrane at 24 cm were tested; the pressure drops were 38.6 and 10.2 mbar, respectively. Therefore, the twin-tower adsorption system with the staggered stainless steel sheet mounted in the towers was set up, as shown in Figure 1, and the carrier was coated with the SiO_2_/MCM41/activated carbon composite membrane. The dimensions of the adsorption tower and the stainless steel carrier were 22 × 22 × 250 mm^3^ and 18 × 18 × 2 mm^3^, respectively. In the process of reforming natural gas to produce hydrogen, methanol reacts with water to obtain products including carbon monoxide, carbon dioxide, and hydrogen, in which carbon monoxide is converted to carbon dioxide by a catalytic converter. The ratio of carbon dioxide to hydrogen is approximately 1:4. Since air must be added into the reforming process as fuel, the exhaust gas of this process may contain carbon dioxide, water vapor, nitrogen, and hydrogen. Therefore, the ratio in the gas phase was set to 60% H_2_, 15% CO_2_, 15% N_2_, and 10% water vapor in this study. The adsorption processes were kept at room temperature at 23~27 °C. Inlet gases were mixed fully before entering the adsorption tower to make the adsorption proceed smoothly in the adsorption tower. Concentrations of H_2_ and N_2_ were detected by a GC-TCD with a MS-5A column, and the concentration of CO_2_ was detected with a Porapak-Q column. A thermometer TES-1160 was used to measure the concentration of water vapor.

## 3. Results and Discussion

### 3.1. Characteristic of the Composite Membrane

The SiO_2_/MCM41/activated carbon composite membrane was characterized by X-ray diffraction (XRD) analysis and scanning electron microscopy (SEM). Figure 2 shows the XRD pattern for the composite membrane prepared by deep coating, and the characteristic peak is found for SiO_2_ at 23° [20]. Since the commercial coconut shell activated carbon was added to the casting solution, the characteristic peak for that was at 24.5°. In addition, the peak at 44.9° showed that the composition of the composite membrane also included diamond [21], and it could be used for the adsorption of volatile organic compounds. The characteristic peak at 2.6° indicated the presence of MCM41 [22]. As mentioned above, to reduce the adsorption of water vapor from molecular sieves, MCM41 was added to the casting solution and the characteristic peak for that was a small-angle X-ray diffraction and a sharp peak in the XRD pattern. Figure 3 shows the cross-sectional image of the SiO_2_/MCM41/activated carbon composite membrane. The supported stainless steel sheet is in the bottom, and the porous membrane can be found in this image. As shown in this image, particles of different sizes seemed to embed in the interior of the gel-like appearance. In addition, uniform distributions of porosity and particles can also be found in the image. As mentioned above, the composition and the porous structure of the composite membrane made it an adsorbent in an adsorber. An elemental composition analysis obtaining using energy dispersive X-ray spectroscopy (EDS) is shown in Figure 4, and the result was obtained from the membrane synthesized by 1.86 mL of TEOS, 7 g of MCM41, and 6 g of activated carbon. The atomic ratio was 71.14: 20.00: 8.01 and the weight ratio was 59.55: 22.32: 15.65 for carbon, oxygen, and silicon in the elemental analysis. The analyzed result contained trace amounts of Cl and Br due to the hydrochloric acid used in preparing the casting solution and a small amount of Br contained in the MCM41. To measure the gas permeability by a gas permeability analyzer (GPA) (Yanaco GTR-10), the membranes were cut into circular samples with a diameter of 3.5 cm. The tested sample was placed into the membrane cell of GAP, and the operating pressure and temperature were controlled at 0.1 MPa and 35 °C, respectively. The downstream side of the membrane was evacuated by a vacuum pump, and then the tested gas was introduced to the upstream side of the membrane. After the permeating process was stable, the amount of the tested gas in the downstream side was collected and measured to obtain the gas permeability. The permeabilities for H_2_, CO_2_, N_2_ and H_2_O were 86.6, 3.6, 5.2, and 4.6 barrer, 10^−10^ cm^3^ (STP) cm/(cm^2^□s□cmHg), and the results showed that the composite membrane was favorable for the purification of hydrogen.

### 3.2. Discussion on Dynamic Curve

The purification of hydrogen from the mixed gas was conducted by Abdeljaoued et al. [23] and Park et al. [24], and the absorber heights were 27.5 and 80 cm, respectively. Although the adsorber height designed by Park et al. [24] was 80 cm, the packed height of the activated carbon and the zeolite 13X was between 5 and 75 cm. Therefore, their breakthrough curves were compared with this study. The separation of hydrogen from the mixture of H_2_, CO_2_, and N_2_ was conducted by Abdeljaoued et al. [23], and the breakthrough curve experienced a decline and then leveled off twice, as shown in Figure 5. The results could be attributed to the fact that the outflow of nitrogen and carbon dioxide led to declines in hydrogen concentration. Similarly, the separation of hydrogen from the mixture of H_2_, CO_2_, CH_4_, and CO was conducted by Park et al. [24], and the breakthrough curve also experienced a decline and then leveled off twice. Since the concentration of CO in the feed mixture is relatively small, the breakthrough curve was not affected by the outflow of CO. Since the feed flow rates were 3.0 L/min and 0.5 L/min in the studies of Abdeljaoued et al. [23] and Park et al. [24], larger than the 0.07 to 0.18 L/min in this study, the time for outflow of pure hydrogen was longer in this study. Although smaller flow rates were used in this study, the time for pure hydrogen outflowing was six to eight times that of the literature data. The results demonstrated that the design of the staggered stainless steel sheet coated with SiO_2_/MCM41/activated carbon composite membrane mounted in the twin-tower adsorption system to purify hydrogen is successful.

For purifying hydrogen in flue gas, water vapor could be removed by alumina and silica gel, CO_2_ and hydrocarbons could be removed by activated carbon, and O_2_, N_2_, CO, and CH_4_ could be removed by molecular sieve. Since the affinity between hydrogen and these adsorbents are poor, hydrogen is vented from the adsorption system for purification. Therefore, the adsorber packed by the staggered stainless steel sheets coated with SiO_2_/MCM41/activated carbon composite membrane was used to purify the hydrogen. When the inlet flow rate increases, the purity of H_2_ at the outlet decreases rapidly with an increase in flow rate, as shown in Figure 6. The reason can be attributed to the fact that an increase in flow rate makes the membrane in the system reach the saturation state more rapidly. Therefore, an outflow of carbon dioxide, nitrogen, and water vapor was found in the outlet of the adsorption tower, causing the ratio of hydrogen concentration to the total concentration of gas decrease rapidly as the flow rate increased. The effect of membrane thickness on adsorption behavior is shown in Figure 7, and the figure shows that the dynamic ratios of concentration of hydrogen to total concentration were almost similar for membrane thicknesses larger than 0.2 cm. It could be deduced from the result that it was difficult for the adsorbate to reach the deeper porosity in the composite membrane, and thus membranes of about 0.2 cm were prepared for testing. Single components adsorbed by the composite membrane in the adsorption system were also tested in this study, as shown in Figure 8. The addition of 6 g activated the 7 g of MCM41 and the 1.86 mL of TEOS in the casting solution and resulted in the breakthrough times 70, 80, and 115 min for H_2_O, N_2_, and CO_2_. This may be the reason for the different flat segments discussed in the following.

### 3.3. Effect of the Amount of Target Adsorbent on Adsorption Behavior

The changes of the ratio of hydrogen concentration to the total concentration of gas affected by an added amount of target adsorbate were shown in Figure 8, Figure 9 and Figure 10. Not only were the adsorption behaviors affected by the amount of target adsorbent but also the suitable amount added to the casting solution could be discussed by these figures. The effect of different amounts of activated carbon on the adsorption behavior was shown in Figure 9, and the results showed that non-hydrogen gas flowed out of the adsorption tower in about 62 min when 5 g of activated carbon was added to the casting solution. The reason could be attributed to the fact that the added activated carbon was not enough, which made the reverse breakthrough curve first drop at this time. When 7 g of activated carbon was added, the slightly flat reverse-breakthrough curve at the ratio of about 0.7 could be found in about 89 min, indicating that this amount was excessive. When 6 g of activated carbon was added, the reverse breakthrough curve began to drop in about 71 min. The results showed that the added amount of activated carbon was appropriate for this preparation. Therefore, 6 g of activated carbon was the better amount added to the casting solution. The effect of different amounts of MCM41 on the adsorption behavior is shown in Figure 10, and the results showed that non-hydrogen gas flowed out of the adsorption tower in about 66 min when 6 g of MCM41 was added to the casting solution. The reason could be attributed to the fact that the added MCM41 was not enough, which made the reverse-breakthrough curve first drop at this time. When 8 g of MCM41 was added, a slightly flat reverse breakthrough curve at the ratio of about 0.7 could be found in about 89 min, indicating that this amount was excessive. When 7 g of MCM41 was added, the reverse breakthrough curve began to drop in about 71 min. The results showed that the added amount of MCM41 was appropriate for this preparation. Therefore, 7 g of MCM41 was the better amount added to the casting solution. Since the ratios of CO_2_ and N_2_ in the gas phase were the same at 15%, the ratio of hydrogen to hydrogen plus non-hydrogen was the same in Figure 8 and Figure 9; the difference was in time for those flat segments. The effect of the different amounts of TEOS on the adsorption behavior is shown in Figure 11, and the results showed that non-hydrogen gas flowed out of the adsorption tower in about 68 min when 1.40 mL of TEOS was added to the casting solution. The reason could be attributed to the fact that the added TEOS was not enough, which made the reverse breakthrough curve first drop at this time. When 2.33 mL of TEOS was added, the slightly flat reverse breakthrough curve at the ratio about 0.66 could be found in about 89 min, indicating that this amount was excessive. When 1.86 mL of TEOS was added, the reverse breakthrough curve began to drop in about 71 min. The results showed that the added amount of TEOS was appropriate for this preparation. Therefore, 1.86 mL TEOS was the better amount added to the casting solution.

### 3.4. Switch Time for Stripping

Regeneration of the saturated adsorption tower could be conducted by the stripping method. To discuss the switching time for the twin-tower adsorption system, the desorption tower, that is, the saturated tower, in the stripping process was heated to 70 °C, and the adsorbate was taken away by the heated air. Figure 7, Figure 8 and Figure 9 show that the outflow of non-hydrogen components could be found at about 66 min, which indicated that the time for the composite membrane reaching the saturated state was about 66 min under the operating conditions in this study. The switching time of the twin-tower adsorption system should be set before 66 min. As shown in Table 1, the required stripping time was 83 min when the adsorbent was nearly saturated at 66 min; the required stripping time was 33 to 43 min when the adsorption tower was operated for 35 to 45 min. Although the stripping time was increased from 40 to 43 min for 40 to 45 min of adsorption, the increased rate was decreased. The results showed that stripping in the saturated state is the more energy-consuming stage. The best switching time is 45 min for this system. The adsorption capacity with 70~80% of the saturated capacity is appropriate for switching to the stripping process. Therefore, the best switching time should be set at the adsorption capacity reaching about 75% of the saturated capacity.

## 4. Conclusions

A twin-tower hydrogen purification system was successfully established in this study. The effects of flow rate and the amount of target adsorbent preparing the composite membrane on the adsorption behavior were discussed to avoid a waste of resources in acquiring hydrogen energy. According to the experimental results, the best operating condition of the twin-tower hydrogen purification system is a lower gas flow rate and a higher composition of the target adsorbent, indicating a longer time for the breakthrough or outflow of hydrogen. The experimental results confirmed that the performance of the hydrogen purification system would not be increased when the amount of one of the target adsorbents was excessive. In contrast, an early outflow of non-hydrogen components could be found when the amount of a certain adsorbent was not sufficient. The experimental results proved that the optimal adsorbent for adding to the casting solution to prepare the composite membrane for use in the twin-tower hydrogen purification system was 1.86 mL of TEOS, 6 g of activated carbon, and 7 g of MCM41. The best switching time was 45 min for this system. The adsorption capacity with 70~80% of the saturated capacity is appropriate for switching to the stripping process. Therefore, the best switching time should be set at an adsorption capacity reaching about 75% of the saturated capacity.

## Figures and Tables

**Figure 1 membranes-11-00169-f001:**
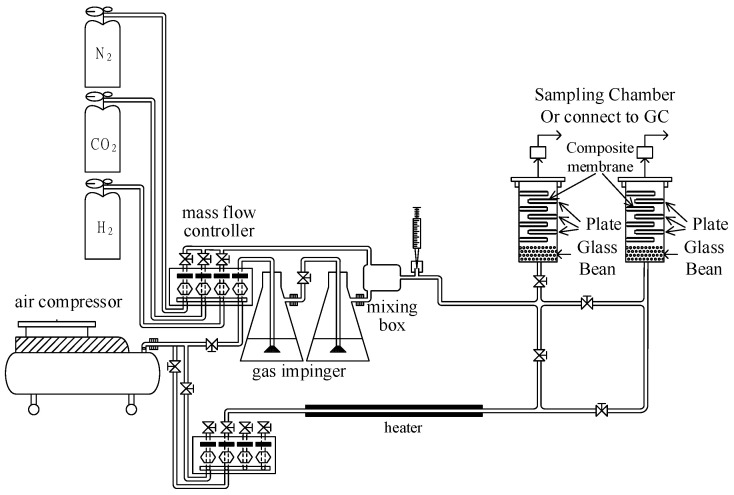
The twin-tower adsorption system in this study.

**Figure 2 membranes-11-00169-f002:**
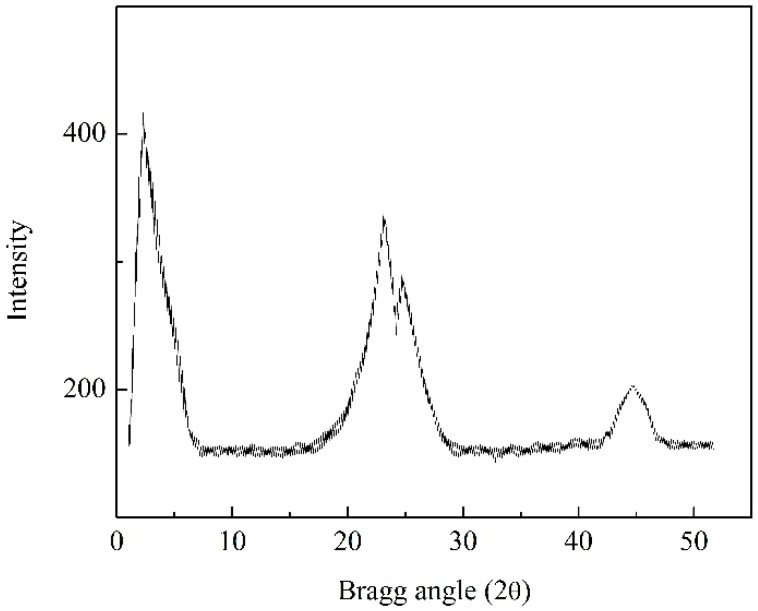
The X-ray diffraction pattern of the composite membrane.

**Figure 3 membranes-11-00169-f003:**
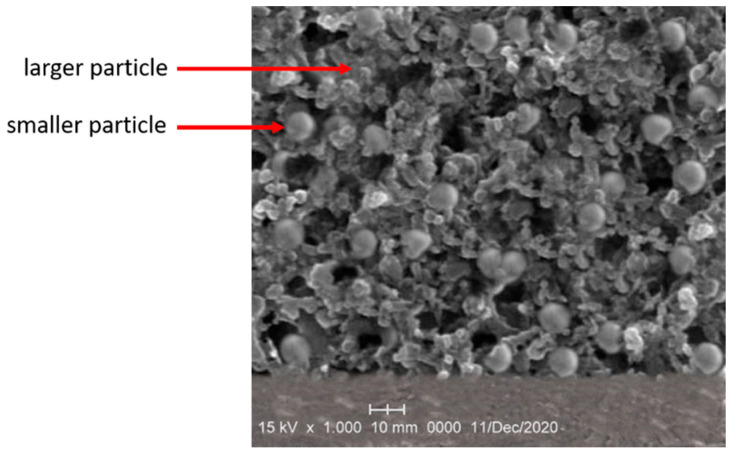
An SEM image of the composite membrane.

**Figure 4 membranes-11-00169-f004:**
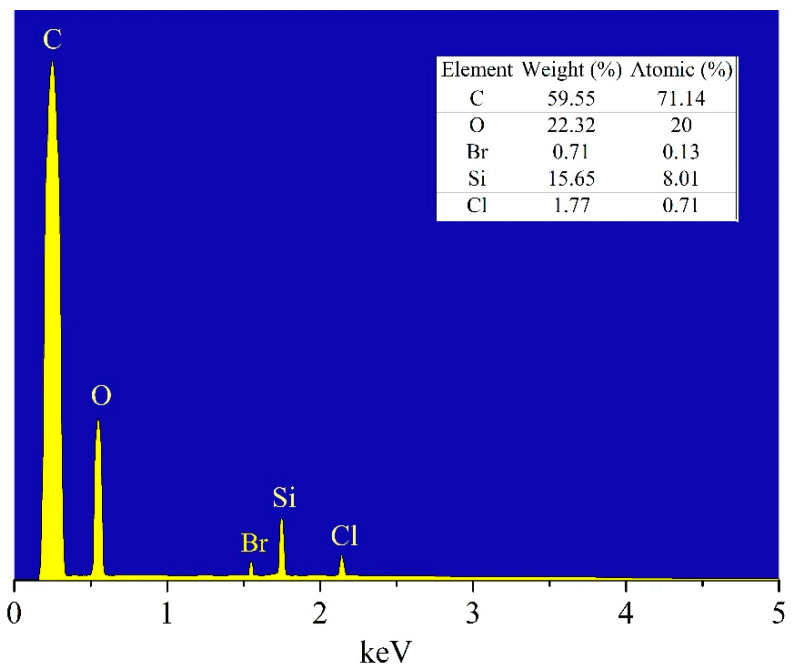
An elemental composition analysis of the composite membrane.

**Figure 5 membranes-11-00169-f005:**
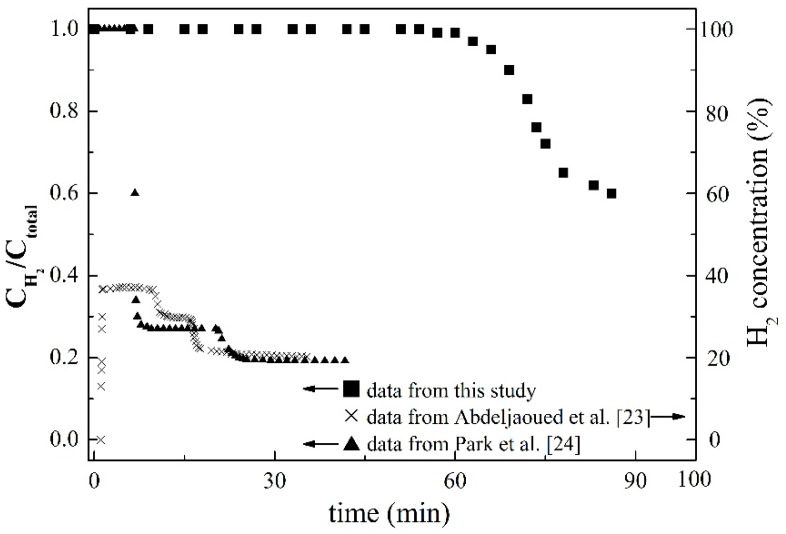
The breakthrough curve compared with literature data.

**Figure 6 membranes-11-00169-f006:**
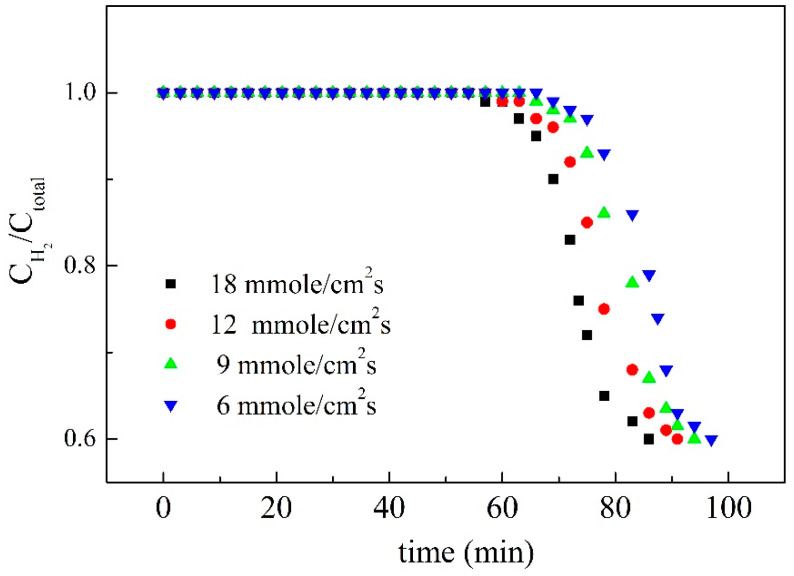
The effect of flow rate on the ratio of hydrogen concentration to the total concentration of gas.

**Figure 7 membranes-11-00169-f007:**
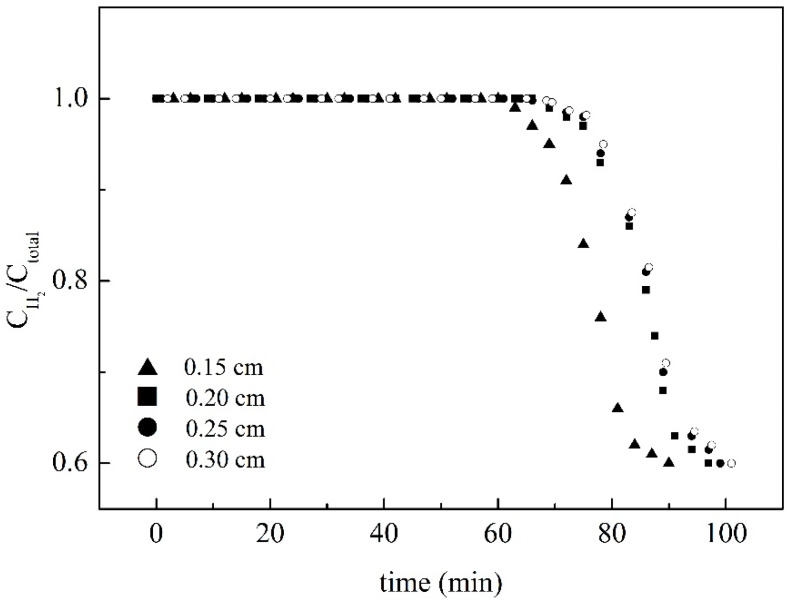
The effect of membrane thickness on adsorption behavior.

**Figure 8 membranes-11-00169-f008:**
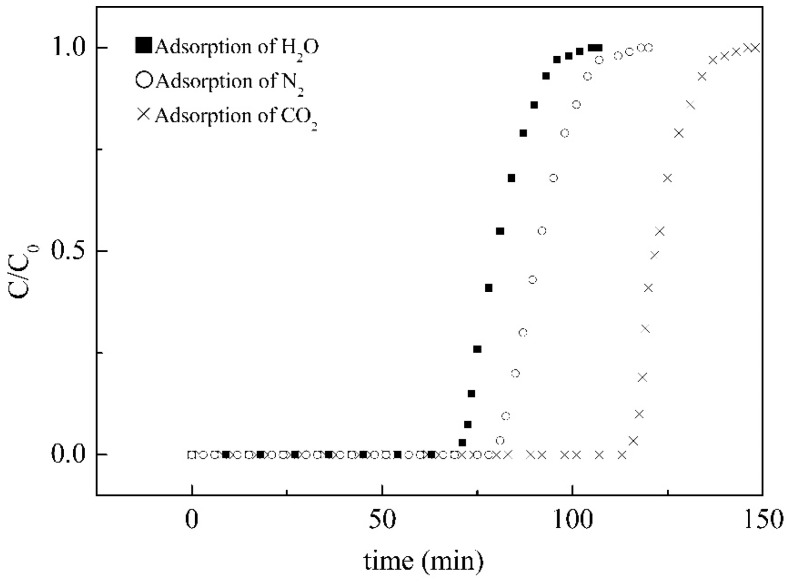
Breakthrough curves for the adsorption of a single component by the composite membrane in the adsorption system.

**Figure 9 membranes-11-00169-f009:**
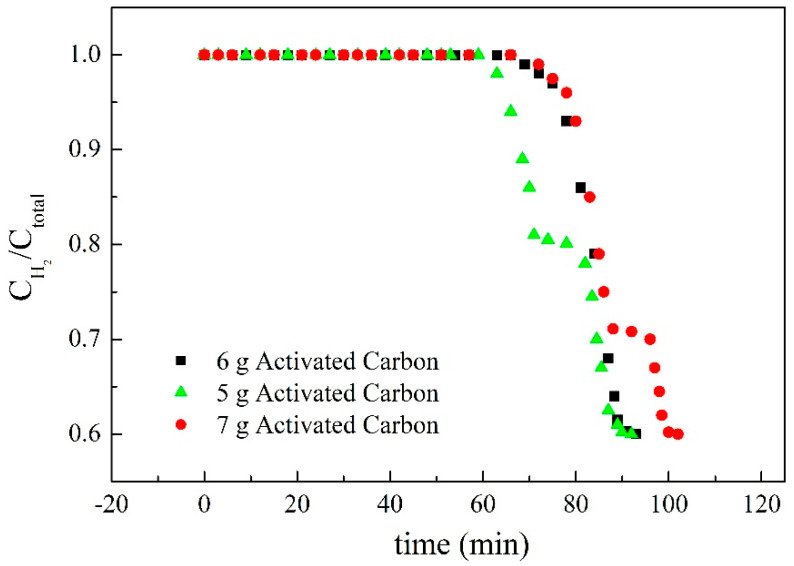
The effect of the addition of activated carbon on the ratio of hydrogen concentration to the total concentration of gas.

**Figure 10 membranes-11-00169-f010:**
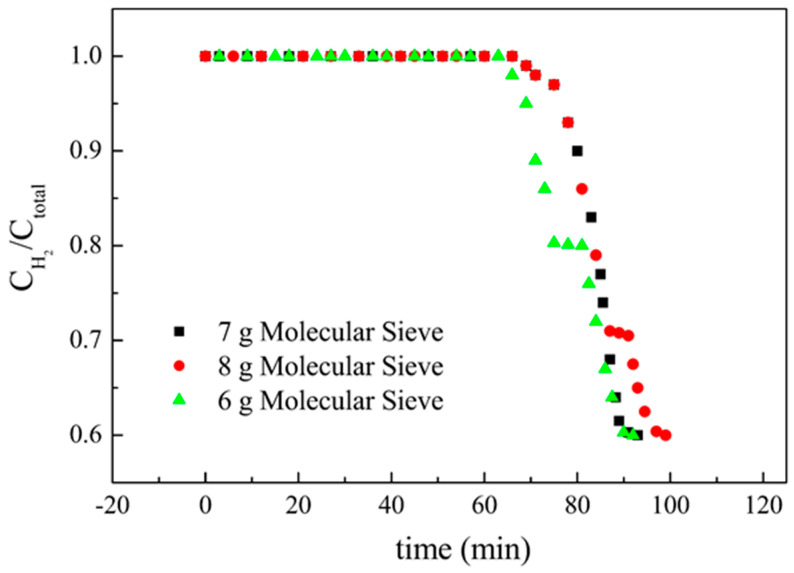
The effect of the addition of molecular sieve on the ratio of hydrogen concentration to the total concentration of gas.

**Figure 11 membranes-11-00169-f011:**
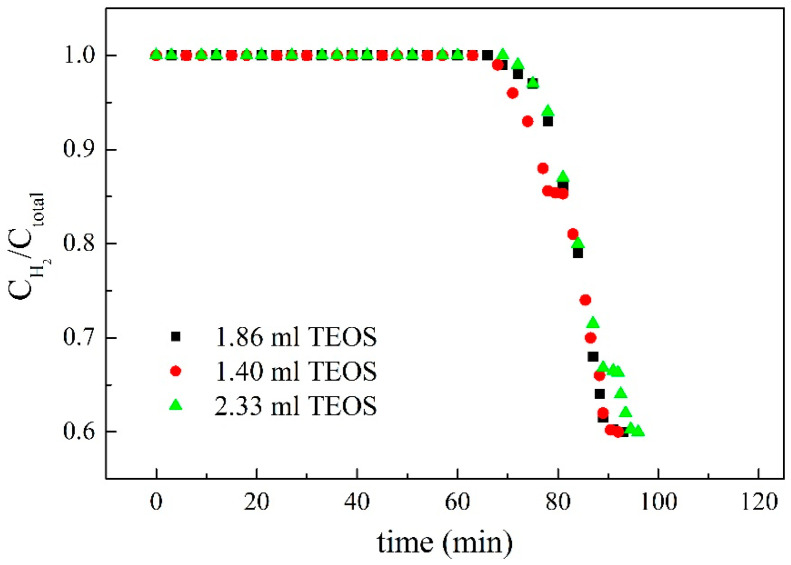
The effect of the addition of tetraethoxysilane (TEOS) on the ratio of hydrogen concentration to the total concentration of gas.

**Table 1 membranes-11-00169-t001:** The relationship between adsorption and stripping time.

Adsorption Time (min)	Stripping Time (min)
35	33
40	40
45	44
50	52
55	65
66	80

## Data Availability

Data sharing is not applicable to this article.

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
