# Peer review of "Adsorption Behaviors of a Twin-Tower Hydrogen Purification System Mounted onto Staggered Stainless Steel Sheets Coated with Composite Membrane"

_membranes, 2021, doi:10.3390/membranes11030169_

Round 1

Reviewer 1 Report

Similar studies have been raised by many scientists for a long time. The number of progressive projects using hydrogen in the field of large and distributed energy, energy storage and all types of transport, from automobiles to airplanes and ships, is becoming "serial". One gets the impression that right now, an electrical revolution is taking place in the world. It can lead to a change in the way of energy systems and the gradual formation of a common global energy market. Taken together, this undoubtedly speaks of the relevance of this article. However, there are the following critical notes on which you need to make adjustments before this article could be recommend for publication.

1. In 2.2. you state that "Comparison between the fixed-bed adsorber and the tower with a number of staggered aluminas coated the composite membrane, the lower pressure drop is for the latter." What is the numerical value of this drop? Also, you are claiming that your system is similar in performance to pressure swing adsorption systems. However, the article lacks any specific numerical values of pressure. The operating pressure range of your system must be specified.

2. There is no data on the operating temperature range of your system. It is necessary to clarify at what temperatures your system operates.

3. The article presents only the data on the dependence of the ratio of the hydrogen concentration to the total gas concentration on time. However, no other data that reflect the hydrogen efficiency of the developed membrane had presented. Data on hydrogen permeability should be added to the article, namely the dependence of the hydrogen flux density through the developed composite membrane on the pressure on the inlet side of the membrane and on temperature.

4. For a full analysis of the developed objects, it is necessary to have data of their elemental composition analysis by energy dispersive X-ray spectroscopy (EDS). It would be liked to see the EDS data in your article.

5. In 3.1 you state that the surface of the composite membrane you have obtained has a gel-like morphology, but you only provide a SEM image for confirmation. This type of research uses atomic force microscopy (AFM) to accurately determine surface morphology. It would be liked to see in your article the AFM data of the surface of your composite membrane.

6. In 3.2 you state that "the dynamic ratios of concentration of hydrogen to total concentration were almost similar for the membrane thickness larger than 200 mm." Does this mean that these characteristics will be similar with a further increase in thickness?

7. Throughout the text of the article, there are a fairly small number of sources (only 21). Also, there are not enough links to relevant sources, therefore many statements seem unfounded.

In its present form, the article is not recommended for publication in the Membranes and requires serious work.

Reviewer 2 Report

I’ve just finished a review of the paper membranes-1112578 titled: “Adsorption Behaviors of a Twin-Tower Hydrogen Purification System Mounted on the Staggered Stainless Steel Sheet Coated with Compsite Membrane” and written by authors: Hung-Ta Wu and Chin-Chun Chung.

In this paper the staggered stainless steel sheet coatings with SiO2/MCM41/activated carbon composite membrane were mounted in the twin-tower adsorption system to purify hydrogen. Similar to the pressure swing adsorption (PSA) system, the amounts of SiO2 and stainless steel sheet, activated carbon, and molecular sieve used in the adsorption tower were changed into the amounts of TEOS, activated carbon powder, and MCM41 powder added to the casting solution. The experimental results showed that the performance of this twin-tower hydrogen purification system would not be increased when one of the target adsorbents was excessive. In addition, the outflow of non-hydrogen components early was found when a certain adsorbent was not enough. The best switching time for this system set at the adsorption capacity reaching about 75% saturated capacity was recommended.

The paper is generally interesting for reading and is in accordance with topic of the journal. It contains and present as well as described interesting results which be useful for application in practice.

 I believe that it should be accepted for publication. However, there are some things which must be improved, and because of that I suggest acceptance but after major revision.

My other comments are given bellow:

  1. Figures should be placed at the place in the text where are mentioned for the first time.
  2. For XRD diagram, how authors can explain presence of diamond? Reference is required.
  3. From authors it is required to insert EDS image and elements mapping together with SEM image? On that way it will be possible to completely define are constituents uniformly or nonuniformly arranged in the membrane. Also, in SEM image authors should note by arrows what are different particles which are visible.
  4. In giving explanations for the results, beside explanations made by themselves, authors also must give explanations which are supported by references. Also, comparison with results from literature is missing.
  5. Also, explanations of purification mechanism(s) are missing.
  6. Explain why using higher amounts of adsorbents do not cause proportional increasing amount(s) of purified hydrogen

Best regards

Reviewer 3 Report

2.1. Section: Authors say “1 g ethanol, and 1.25 g”. What is the purpose of expressing the results in these units? They should be expressed as mL and / or L. Same with TEOS later in the manuscript. Why pH 2 was used?

Figure 3 description is very poor.

Figure 4: numbers of the right y-axis are missingIn addition, the figure could be improved?

Please, provide the x-axis of the figures from 0 to maximum value (approximately)

Round 2

Reviewer 1 Report

Accept in present form

Reviewer 2 Report

I’ve finished a second review of the paper membranes-1112578.

I would like to thank the authors for their effort to add additional required explanations.

I think that the paper is minimal improved now, but it is enough to satisfy minimal criteria to bi accepted for publication in presented form.

However, I also would like to draw the attention of the authors that in the future they must give more concrete and much more detailed answers and comments.

In particular, in the future, insufficient comparison of results with those that already exist and clear emphasis on the advantages and disadvantages of obtained in relation to what is already known will not be accepted.

Reviewer 3 Report

Accept in present form